# Exploration Based Language Learning for Text-Based Games

## Abstract

This work presents an exploration and imitation-learning-based agent capable of state-of-the-art performance in playing text-based computer games. Text-based computer games describe their world to the player through natural language and expect the player to interact with the game using text. These games are of interest as they can be seen as a testbed for language understanding, problem-solving, and language generation by artificial agents. Moreover, they provide a learning environment in which these skills can be acquired through interactions with an environment rather than using fixed corpora. One aspect that makes these games particularly challenging for learning agents is the combinatorially large action space. Existing methods for solving text-based games are limited to games that are either very simple or have an action space restricted to a predetermined set of admissible actions. In this work, we propose to use the exploration approach of Go-Explore (Ecoffet et al., 2019) for solving text-based games. More specifically, in an initial exploration phase, we first extract trajectories with high rewards, after which we train a policy to solve the game by imitating these trajectories. Our experiments show that this approach outperforms existing solutions in solving text-based games, and it is more sample efficient in terms of the number of interactions with the environment. Moreover, we show that the learned policy can generalize better than existing solutions to unseen games without using any restriction on the action space.

## 1 Introduction

Text-based games became popular in the mid 80s with the game series Zork (Anderson & Galley, 1985) resulting in many different text-based games being produced and published (Spaceman, 2019). These games use a plain text description of the environment and the player has to interact with them by writing natural-language commands. Recently, there has been a growing interest in developing agents that can automatically solve text-based games (Côté et al., 2018) by interacting with them. These settings challenge the ability of an artificial agent to understand natural language, common sense knowledge, and to develop the ability to interact with environments using language (Luketina et al., 2019; Branavan et al., 2012).

Since the actions in these games are commands that are in natural language form, the major obstacle is the extremely large action space of the agent, which leads to a combinatorially large exploration problem. In fact, with a vocabulary of $N$ words (e.g. 20K) and the possibility of producing sentences with at most $m$ words (e.g. 7 words), the total number of actions is $O(N^m)$ (e.g. $20K^7 \approx 1.28e^{30}$). To avoid this large action space, several existing solutions focus on simpler text-based games with very small vocabularies where the action space is constrained to verb-object pairs (DePristo & Zubek, 2001; Narasimhan et al., 2015; Yuan et al., 2018; Zelinka, 2018). Moreover, many existing works rely on using predetermined sets of admissible actions (He et al., 2015; Tessler et al., 2019; Zahavy et al., 2018). However, a more ideal, and still under explored, alternative would be an agent that can operate in the full, unconstrained action space of natural language that can systematically generalize to new text-based games with no or few interactions with the environment.

To address this challenge, we propose to use the idea behind the recently proposed Go-Explore (Ecoffet et al., 2019) algorithm. Specifically, we propose to first extract high reward trajectories of states and actions in the game using the exploration methodology proposed in Go-Explore

and then train a policy using a Seq2Seq (Sutskever et al., 2014) model that maps observations to actions, in an imitation learning fashion. To show the effectiveness of our proposed methodology, we first benchmark the exploration ability of Go-Explore on the family of text-based games called CoinCollector (Yuan et al., 2018). Then we use the 4,440 games of "First TextWorld Problems" (Côté, 2018), which are generated using the machinery introduced by Côté et al. (2018), to show the generalization ability of our proposed methodology. In the former experiment we show that Go-Explore finds winning trajectories faster than existing solutions, and in the latter, we show that training a Seq2Seq model on the trajectories found by Go-Explore results in stronger generalization, as suggested by the stronger performance on unseen games, compared to existing competitive baselines (He et al., 2015; Narasimhan et al., 2015).

**Reinforcement Learning Based Approaches for Text-Based Games**   Among reinforcement learning based efforts to solve text-based games two approaches are prominent. The first approach assumes an action as a sentence of a fixed number of words, and associates a separate $Q$-function (Watkins, 1989; Mnih et al., 2015) with each word position in this sentence. This method was demonstrated with two-word sentences consisting of a verb-object pair (e.g. take apple) (De-Pristo & Zubek, 2001; Narasimhan et al., 2015; Yuan et al., 2018; Zelinka, 2018; Fulda et al., 2017). In the second approach, one $Q$-function that scores all possible actions (i.e. sentences) is learned and used to play the game (He et al., 2015; Tessler et al., 2019; Zahavy et al., 2018). The first approach is quite limiting since a fixed number of words must be selected in advance and no temporal dependency is enforced between words (e.g. lack of language modelling). In the second approach, on the other hand, the number of possible actions can become exponentially large if the admissible actions (a predetermined low cardinality set of actions that the agent can take) are not provided to the agent. A possible solution to this issue has been proposed by Tao et al. (2018), where a hierarchical pointer-generator is used to first produce the set of admissible actions given the observation, and subsequently one element of this set is chosen as the action for that observation. However, in our experiments we show that even in settings where the true set of admissible actions is provided by the environment, a $Q$-scorer (He et al., 2015) does not generalize well in our setting (Section 5.2 Zero-Shot) and we would expect performance to degrade even further if the admissible actions were generated by a separate model. Less common are models that either learn to reduce a large set of actions into a smaller set of admissible actions by eliminating actions (Zahavy et al., 2018) or by compressing them in a latent space (Tessler et al., 2019).

**Exploration in Reinforcement Learning**   In most text-based games rewards are sparse, since the size of the action space makes the probability of observing a reward extremely low when taking only random actions. Sparse reward environments are particularly challenging for reinforcement learning as they require longer term planning. Many exploration based solutions have been proposed to address the challenges associated with reward sparsity. Among these exploration approaches are novelty search (Lehman & Stanley, 2008; 2011; Conti et al., 2018; Achiam & Sastry, 2017; Burda et al., 2018), intrinsic motivation (Schmidhuber, 1991b; Oudeyer & Kaplan, 2009; Barto, 2013), and curiosity based rewards (Schmidhuber, 2006; 1991a; Pathak et al., 2017). For text based games exploration methods have been studied by Yuan et al. (2018), where the authors showed the effectiveness of the episodic discovery bonus (Gershman & Daw, 2017) in environments with sparse rewards. This exploration method can only be applied in games with very small action and state spaces, since their counting methods rely on the state in its explicit raw form.

## 2   METHODOLOGY

Go-Explore (Ecoffet et al., 2019) differs from the exploration-based algorithms discussed above in that it explicitly keeps track of under-explored areas of the state space and in that it utilizes the determinism of the simulator in order to return to those states, allowing it to explore sparse-reward environments in a sample efficient way (see Ecoffet et al. (2019) as well as section 4.1). For the experiments in this paper we mainly focus on the final performance of our policy, not how that policy is trained, thus making Go-Explore a suitable algorithm for our experiments. Go-Explore is composed of two phases. In *phase 1* (also referred to as the "exploration" phase) the algorithm explores the state space through keeping track of previously visited states by maintaining an *archive*. During this phase, instead of resuming the exploration from scratch, the algorithm starts exploring from promising states in the archive to find high performing trajectories. In *phase 2* (also referred

Table 1: Example of the observations provided by the CookingWorld environment.

| *Description* | -= garden = - well, here we are in a garden. There is a roasted red apple and a red onion on the floor | *Admissible commands* |
|---|---|---|
| *Inventory* | You are carrying: a black pepper | - drop black pepper |
| *Prev. Action* | Open screen door | - eat black pepper |
| *Quest* | Gather all following ingredients and follow the directions to prepare this tasty meal. Ingredients: black pepper, red apple salt. Directions: chop the red apple, roast the red apple, prepare meal | - examine red apple - examine red onion - go north - look - take red apple |
| *Feedback* | That 's already open | - take red onion |

to as the "robustification" phase, while in our variant we will call it "generalization") the algorithm trains a policy using the trajectories found in phase 1. Following this framework, which is also shown in Figure 3 (Appendix A.2), we define the Go-Explore phases for text-based games.

Let us first define text-based games using the same notation as Yuan et al. (2018). A text-based game can be framed as a discrete-time Partially Observable Markov Decision Process (POMDP) (Kaelbling et al., 1998) defined by $(S, T, A, \Omega, O, R)$, where: $S$ is the set of the environment states, $T$ is the state transition function that defines the next state probability, i.e. $T(s_{t+1}|a_t; s_t)\forall s_t \in S$, $A$ is the set of actions, which in our case is all the possible sequences of tokens, $\Omega$ is the set of observations, i.e. text observed by the agent every time has it to take an action in the game (i.e. dialogue turn) which is controlled by the conditional observation probability $O$, i.e. $O(o_t|s_t, a_{t-1})$, and, finally, $R$ is the reward function i.e. $r = R(s, a)$.

Let us also define the observation $o_t \in \Omega$ and the action $a_t \in A$. Text-based games provide some information in plain text at each turn and, without loss of generality, we define an observation $o_t$ as the sequence of tokens $\{o_t^0, \cdots, o_t^n\}$ that form that text. Similarly, we define the tokens of an action $a_t$ as the sequence $\{a_t^0, \cdots, a_t^m\}$. Furthermore, we define the set of admissible actions $\mathcal{A}_t \in A$ as $\mathcal{A}_t = \{a_0, \cdots, a_z\}$, where each $a_i$, which is a sequence of tokens, is grammatically correct and admissible with reference to the observation $o_t$.

## 2.1 PHASE 1: EXPLORATION

In phase 1, Go-Explore builds an archive of *cells*, where a cell is defined as a set of observations that are mapped to the same, discrete representation by some mapping function $f(x)$. Each cell is associated with meta-data including the trajectory towards that cell, the length of that trajectory, and the cumulative reward of that trajectory. New cells are added to the archive when they are encountered in the environment, and existing cells are updated with new meta-data when the trajectory towards that cells is higher scoring or equal scoring but shorter.

At each iteration the algorithm selects a cell from this archive based on meta-data of the cell (e.g. the accumulated reward, etc.) and starts to randomly explore from the end of the trajectory associated with the selected cell. Phase 1 requires three components: the way that observations are embedded into cell representations, the cell selection, and the way actions are randomly selected when exploring from a selected cell. In our variant of the algorithm, $f(x)$ is defined as follows: given an observation, we compute the word embedding for each token in this observation, sum these embeddings, and then concatenate this sum with the current cumulative reward to construct the cell representation. The resulting vectors are subsequently compressed and discretized by binning them in order to map similar observations to the same cell. This way, the cell representation, which is the key of the archive, incorporates information about the current observation of the game. Adding the current cumulative reward to the cell representation is new to our Go-Explore variant as in the original algorithm only down-scaled image pixels were used. It turned out to be a very very effective to increase the speed at which high reward trajectories are discovered. In phase 1, we restrict the action space to the set of admissible actions $\mathcal{A}_t$ that are provided by the game at every step of the game [1]. This too is particularly important for the random search to find a high reward trajectory faster. Finally, we denote the trajectory found in phase 1 for game $g$ as $\mathcal{T}_g = [(o_0, a_0, r_0), \cdots, (o_t, a_t, r_t)]$.

---

[1]Note that the final goal is to generalize to test environments where admissible actions are not available. The assumption that admissible actions are available at training time holds in cases where we build the training environment for an RL agent (e.g. a hand-crafted dialogue system), and a system trained in such an environment can be practically applied as long as the system does not rely on such information at test time. Thus, we assumed that these admissible commands are not available at test time.

Table 2: Statistics of the two families of text-based games used in the experiments. The average is among the different games in CookingWorld and among different instance of the same game for CoinCollector.

|  | CoinCollector | CookingWorld |
|---|---|---|
| Vocabulary size ($|V|$) | 1,250 | 20,000 |
| Action Space | 10 | 20,000[5] |
| #Level | 1 (Hard) | 222 |
| Max Room | 90 | 12 |
| Avg. #Tok. Description | $64 \pm 9$ | $97 \pm 49$ |
| Avg. #Adm. Actions | 10 | $14 \pm 13$ |

## 2.2 PHASE 2: GENERALIZATION

Phase 2 of Go-Explore uses the trajectories found in phase 1 and trains a policy based on those trajectories. The goal of this phase in the original Go-Explore algorithm is to turn the fragile policy of playing a trajectory of actions in sequence into a more robust, state-conditioned policy that can thus deal with environmental stochasticity. In our variant of the algorithm the purpose of the second phase is generalization: although in our environment there is no stochasticity, our goal is to learn a general policy that can be applied across different games and generalize to unseen games. In the original Go-Explore implementation, the authors used the backward Proximal Policy Optimization algorithm (PPO) (Salimans & Chen, 2018; Schulman et al., 2017) to train this policy. In this work we opt for a simple but effective Seq2Seq imitation learning approach that does not use the reward directly in the loss. More specifically, given the trajectory $\mathcal{T}_g$[2], we train a Seq2Seq model to minimize the negative log-likelihood of the action $a_t$ given the observation $o_t$. In other words, consider a word embedding matrix $E \in \mathbb{R}^{d \times |V|}$ where $d$ is the embedding size and $|V|$ is the cardinality of the vocabulary, which maps the input token to an embedded vector. Then, we define an encoder $\text{LSTM}_{enc}$ and a decoder $\text{LSTM}_{dec}$. Every token $o_t$ from the trajectory $\mathcal{T}_g$ is converted to its embedded representation using the embedding matrix $E$ and the sequence of these embedding vectors is passed through $\text{LSTM}_{enc}$:

$$h_i^{enc} = \text{LSTM}_{enc}(E(o_t^i), h_{i-1}^{enc}). \tag{1}$$

The last hidden state $h_{|o_t|}^{enc}$ is used as the initial hidden state of the decoder which generates the action $a_t$ token by token. Specifically, given the sequence of hidden states $H \in \mathbb{R}^{d \times |o_t|}$ of the encoder, tokens $a_t^j$ are generated as follows:

$$h_j^{dec} = \text{LSTM}_{dec}(E(a_t^{(j-1)}), h_{j-1}^{dec}) \tag{2}$$

$$c_j = \text{Softmax}(h_t^{dec} H) H^T \tag{3}$$

$$dist_t^j = \text{Softmax}(W[h_j^{dec}; c_j]) \tag{4}$$

where $W \in \mathbb{R}^{2d \times |V|}$ is a matrix that maps the decoder hidden state, concatenated with the context vector, into a vocabulary size vector. During training, the parameters of the model are trained by minimizing:

$$L_{P(a_t|o_t)} = -\sum_k^{|a_t|} \log(dist_k^j(a_t)) \tag{5}$$

which is the sum of the negative log likelihood of each token in $a_t$ (using teacher forcing (Williams & Zipser, 1989)). However, at test time the model produces the sequence in an auto-regressive way (Graves, 2013) using greedy search.

## 3 EXPERIMENTS

### 3.1 GAMES AND EXPERIMENTS SETUP

A set of commonly used standard benchmarks (Yuan et al., 2018; Côté et al., 2018; Narasimhan et al., 2015) for agents that play text-based games are simple games which require no more than two words in each step to solve the game, and have a very limited number of admissible actions

---

[2]Or the set of trajectories, depending on the training setting

Figure 1: CoinCollector results of DQN++ and DRQN++ (Yuan et al., 2018) versus Go-Explore Phase1, i.e. just exploration.

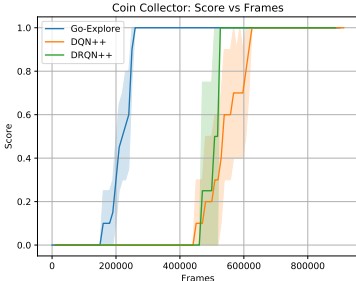

| Model | Frames | Steps |
|---|---|---|
| *DQN++* | $546.74K \pm 55.75K$ | $38.8 \pm 0.2$ |
| *DRQN++* | $508.96K \pm 23.61K$ | $42.0 \pm 0.8$ |
| *Go-Phase1* | $205.00K \pm 29.41K$ | $30 \pm 0.0$ |

per observation. While simplifying, this setting limits the agent's ability to fully express natural language and learn more complex ways to speak. In this paper, we embrace more challenging environments where multiple words are needed at each step to solve the games and the reward is particularly sparse. Hence, we have selected the following environments:

- **CoinCollector** (Yuan et al., 2018) is a class of text-based games where the objective is to find and collect a coin from a specific location in a given set of connected rooms [3]. The agent wins the game after it collects the coin, at which point (for the first and only time) a reward of +1 is received by the agent. The environment parses only five admissible commands (go north, go east, go south, go west, and take coin) made by two worlds;

- **CookingWorld**[4] (Côté, 2018) in this challenge, there are 4,440 games with 222 different levels of difficulty, with 20 games per level of difficulty, each with different entities and maps. The goal of each game is to cook and eat food from a given recipe, which includes the task of collecting ingredients (e.g. tomato, potato, etc.), objects (e.g. knife), and processing them according to the recipe (e.g. cook potato, slice tomato, etc.). The parser of each game accepts 18 verbs and 51 entities with a predefined grammar, but the overall size of the vocabulary of the observations is 20,000. In Appendix A.1 we provide more details about the levels and the games' grammar.

In our experiments, we try to address two major research questions. First, we want to benchmark the exploration power of phase 1 of Go-Explore in comparison to existing exploration approaches used in text-based games. For this purpose, we generate 10 CoinCollector games with the hardest setting used by Yuan et al. (2018), i.e. hard-level 30 (refer to the Appendix A.1 for more information) and use them as a benchmark. In fact, CoinCollector requires many actions (at least 30 on hard games) to find a reward, which makes it suitable for testing the exploration capabilities of different algorithms. Secondly, we want to verify the generalization ability of our model in creating complex strategies using natural language. CoinCollector has a very limited action space, and is mainly designed to benchmark models on their capability of dealing with sparse rewards. Therefore we use the more complex CookingWorld games to evaluate the generalization capabilities of our proposed approach. We design three different settings for CookingWorld: 1) *Single:* treat each game independently, which means we train and test one agent for each game to evaluate how robust different models are across different games.; 2) *Joint:* training and testing a single policy on all the 4,440 CookingWorld games at the same time to verify that models can learn to play multiple games at the same time; 3) *Zero-Shot:* split the games into training, validation, and test sets, and then train our policy on the training games and test it on the unseen test games. This setting is the hardest among all, since it requires generalization to unseen games.

In both CoinCollector and CookingWorld games an observation $o_t$ provided by the environment consists of a room description $D$, inventory information $I$, quest $Q$, previous action $P$ and feedback $F$ provided in the previous turn. Table 1 shows an example for each of these components. In our experiments for phase 1 of Go-Explore we only use $D$ as the observation.

---

[3] Visual example provided at: https://github.com/xingdi-eric-yuan/TextWorld-CoinCollector

[4] We gave this name to this family of games, following the naming convention of TextWorld environments

## 3.2 BASELINES

For the CoinCollector games, we compared Go-Explore with the episodic discovery bonus (Gershman & Daw, 2017) that was used by Yuan et al. (2018) to improve two Q-learning-based baselines: DQN++ and DRQN++. We used the code provided by the authors and the same hyper-parameters[5].

For the CookingWorld games, we implemented three different treatments based on two existing methods:

- **LSTM-DQN** (Narasimhan et al., 2015; Yuan et al., 2018): An LSTM based state encoder with a separate $Q$-functions for each component (word) of a fixed pattern of Verb, Adjective1, Noun1, Adjective2, and Noun2. In this approach, given the observation $o_t$, the tokens are first converted into embeddings, then an LSTM is used to extract a sequence of hidden states $H_{dqn} \in \mathbb{R}^{d \times |o_t|}$. A mean-pool layer is applied to $H_{dqn}$ to produce a single vector $h_{o^t}$ that represents the whole sequence. Next, a linear transformation $W_{type} \in \mathbb{R}^{d \times |V_{type}|}$ is used to generate each of the Q values, where $|V_{type}| \ll |V|$ is the subset of the original vocabulary restricted to the word type of a particular game (e.g for Verb type: take, drop, etc.). Formally, we have:

$$Q(o_t, a_{type}) = h_{o^t} W_{type} \qquad \text{where, type} \in \{\text{Verb, Obj, Noun, Obj2, Noun2}\}. \qquad (6)$$

  Next, all the $Q$-functions are jointly trained using the DQN algorithm with $\epsilon$-greedy exploration (Watkins, 1989; Mnih et al., 2015). At evaluation time, the argmax of each $Q$-function is concatenated to produce $a_t$. Importantly, in $V_{type}$ a special token $$ is used to denote the absence of a word, so the model can produce actions with different lengths. Figure 4 in Appendix A.2 shows a depiction of this model.

- **LSTM-DQN+ADM:** It is the same model as LSTM-DQN, except that the random actions for $\epsilon$-greedy exploration are sampled from the set of admissible actions instead of creating them by sampling each word separately.

- **DRRN** (He et al., 2015): In this approach a model learns how to score admissible actions instead of directly generating the action token by token. The policy uses an LSTM for encoding the observation and actions are represented as the sum of the embedding of the word tokens they contain. Then, the $Q$ value is defined as the dot product between the embedded representations of the observation and the action. Following the aforementioned notation, $h_{o^t}$ is generated as in the LSTM-DQN baseline. Next, we define its embedded representation as $c_i = \sum_{k}^{|a_i|} E(a_i^k)$, where $E$ is an embedding matrix as in Equation 1. Thus, the $Q$-function is defined as:

$$Q(o_t, a_i) = h_{o^t} c_i \qquad (7)$$

  At testing time the action with the highest $Q$ value is chosen. Figure 6 in Appendix A.2 shows a depiction of this model.

## 3.3 HYPER-PARAMETERS

In all the games the maximum number of steps has been set to 50. As mentioned earlier, the cell representation used in the Go-Explore archive is computed by the sum of embeddings of the room description tokens concatenated with the current cumulative reward. The sum of embeddings is computed using 50 dimensional pre-trained GloVe (Pennington et al., 2014) vectors. In the Coin-Collector baselines we use the same hyper-parameters as in the original paper. In CookingWorld all the baselines use pre-trained GloVe of dimension 100 for the single setting and 300 for the joint one. The LSTM hidden state has been set to 300 for all the models.

## 4 RESULTS

### 4.1 COINCOLLECTOR

In this setting, we compare the number of actions played in the environment (frames) and the score achieved by the agent (i.e. +1 reward if the coin is collected). In Go-Explore we also count the actions used to restore the environment to a selected cell, i.e. to bring the agent to the state represented

---

[5]The authors released their code at https://github.com/xingdi-eric-yuan/TextWorld-Coin-Collector.

Table 3: CookingWorld results on the three evaluated settings single, joint and zero-shot.

| Model | Single | | | Joint | | | Zero-Shot | | |
|---|---|---|---|---|---|---|---|---|---|
| | *Score* | *Steps* | *Win* | *Score* | *Steps* | *Win* | *Score* | *Steps* | *Win* |
| *LSTM-DQN* | 2206 | 201832 | 412 | 473 | 213618 | 172 | 31 | 21480 | 14 |
| *+ADM* | 10360 | 140940 | 1770 | 623 | 210283 | 200 | 37 | 21530 | 14 |
| *DRRN* | 16075 | 78856 | 3195 | 4560 | 184888 | 216 | 451 | 17243 | 37 |
| *Go-Explore Seq2Seq* | **17507** | 68571 | **3757** | **11167** | 85967 | **2326** | **1038** | 10805 | **207** |
| *Go-Explore Phase 1* | 19437 | 49011 | 4279 | - | - | - | - | - | - |
| *Max Possible* | 19882 | - | 4440 | 19882 | - | 4440 | 2034 | - | 444 |

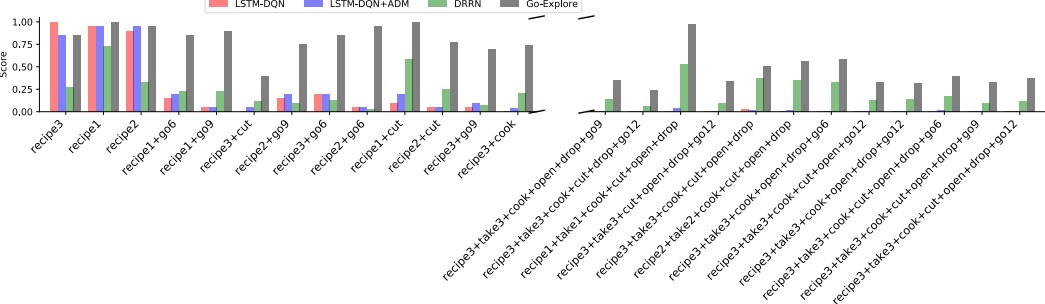

Figure 2: Breakdown of results for the CookingWorld games in the Joint setting. The results are normalized and sorted by increasing difficulty level from left to right, averaged among the 20 games of each level.

in the selected cell. This allows a one-to-one comparison of the exploration efficiency between Go-Explore and algorithms that use a count-based reward in text-based games. Importantly, Yuan et al. (2018) showed that DQN and DRQN, without such counting rewards, could never find a successful trajectory in hard games such as the ones used in our experiments. Figure 1 shows the number of interactions with the environment (frames) versus the maximum score obtained, averaged over 10 games of the same difficulty. As shown by Yuan et al. (2018), DRQN++ finds a trajectory with the maximum score faster than to DQN++. On the other hand, phase 1 of Go-Explore finds an optimal trajectory with approximately half the interactions with the environment. Moreover, the trajectory length found by Go-Explore is always optimal (i.e. 30 steps) whereas both DQN++ and DRQN++ have an average length of 38 and 42 respectively.

## 4.2 COOKINGWORLD

In CookingWorld, we compared models in the three settings mentioned earlier, namely, single, joint, and zero-shot. In all experiments, we measured the sum of the final scores of all the games and their trajectory length (number of steps). Table 3 summarizes the results in these three settings. Phase 1 of Go-Explore on single games achieves a total score of 19,530 (sum over all games), which is very close to the maximum possible points (i.e. 19,882), with 47,562 steps. A winning trajectory was found in 4,279 out of the total of 4,440 games. This result confirms again that the exploration strategy of Go-Explore is effective in text-based games. Next, we evaluate the effectiveness and the generalization ability of the simple imitation learning policy trained using the extracted trajectories in phase 1 of Go-Explore in the three settings mentioned above.

**Single** In this setting, each model is trained from scratch in each of the 4,440 games based on the trajectory found in phase 1 of Go-Explore (previous step). As shown in Table 3, the LSTM-DQN (Narasimhan et al., 2015; Yuan et al., 2018) approach without the use of admissible actions performs poorly. One explanation for this could be that it is difficult for this model to explore both language and game strategy at the same time; it is hard for the model to find a reward signal before it has learned to model language, since almost none of its actions will be admissible, and those reward signals are what is necessary in order to learn the language model. As we see in Table 3, however, by using the admissible actions in the $\epsilon$-greedy step the score achieved by the LSTM-DQN increases dramatically (+ADM row in Table 3). DRRN (He et al., 2015) achieves a very high score, since it explicitly learns how to rank admissible actions (i.e. a much simpler task than generating

text). Finally, our approach of using a Seq2Seq model trained on the single trajectory provided by phase 1 of Go-Explore achieves the highest score among all the methods, even though we do not use admissible actions in this phase. However, in this experiment the Seq2Seq model cannot perfectly replicate the provided trajectory and the total score that it achieves is in fact 9.4% lower compared to the total score achieved by phase 1 of Go-Explore. Figure 7 (in Appendix A.3) shows the score breakdown for each level and model, where we can see that the gap between our model and other methods increases as the games become harder in terms of skills needed.

**Joint**    In this setting, a single model is trained on all the games at the same time, to test whether one agent can learn to play multiple games. Overall, as expected, all the evaluated models achieved a lower performance compared to the single game setting. One reason for this could be that learning multiple games at the same time leads to a situation where the agent encounters similar observations in different games, and the correct action to take in different games may be different. Furthermore, it is important to note that the order in which games are presented greatly affects the performance of LSTM-DQN and DRRN. In our experiments, we tried both an easy-to-hard curriculum (i.e. sorting the games by increasing level of difficulty) and a shuffled curriculum. Shuffling the games at each epoch resulted in far better performance, thus we only report the latter. In Figure 2 we show the score breakdown, and we can see that all the baselines quickly fail, even for easier games.

**Zero-Shot**    In this setting the 4,440 games are split into training, validation, and test games. The split is done randomly but in a way that different difficulty levels (recipes 1, 2 and 3), are represented with equal ratios in all the 3 splits, i.e. stratified by difficulty. As shown in Table 3, the zero-shot performance of the RL baselines is poor, which could be attributed to the same reasons why RL baselines under-perform in the Joint case. Especially interesting is that the performance of DRRN is substantially lower than that of the Go-Explore Seq2Seq model, even though the DRRN model has access to the admissible actions at test time, while the Seq2Seq model (as well as the LSTM-DQN model) has to construct actions token-by-token from the entire vocabulary of 20,000 tokens. On the other hand, Go-Explore Seq2Seq shows promising results by solving almost half of the unseen games. Figure 8 (in Appendix A.3) shows that most of the lost games are in the hardest set, where a very long sequence of actions is required for winning the game. These results demonstrate both the relative effectiveness of training a Seq2Seq model on Go-Explore trajectories, but they also indicate that additional effort needed for designing reinforcement learning algorithms that effectively generalize to unseen games.

## 5    DISCUSSION

Experimental results show that our proposed Go-Explore exploration strategy is a viable methodology for extracting high-performing trajectories in text-based games. This method allows us to train supervised models that can outperform existing models in the experimental settings that we study. Finally, there are still several challenges and limitations that both our methodology and previous solutions do not fully address yet. For instance:

**State Representation**    The state representation is the main limitation of our proposed imitation learning model. In fact, by examining the observations provided in different games, we notice a large overlap in the descriptions ($D$) of the games. This overlap leads to a situation where the policy receives very similar observations, but is expected to imitate two different actions. This show especially in the joint setting of CookingWorld, where the 222 games are repeated 20 times with different entities and room maps. In this work, we opted for a simple Seq2Seq model for our policy, since our goal is to show the effectiveness of our proposed exploration methods. However, a more complex Hierarchical-Seq2Seq model (Sordoni et al., 2015) or a better encoder representation based on knowledge graphs (Ammanabrolu & Riedl, 2019a;b) would likely improve the of performance this approach.

**Language Based Exploration**    In Go-Explore, the given admissible actions are used during random exploration. However, in more complex games, e.g. Zork I and in general the Z-Machine games, these admissible actions are not provided. In such settings, the action space would explode in size, and thus Go-Explore, even with an appropriate cell representation, would have a hard time

finding good trajectories. To address this issue one could leverage general language models to produce a set of grammatically correct actions. Alternatively one could iteratively learn a policy to sample actions, while exploring with Go-Explore. Both strategies are viable, and a comparison is left to future work.

It is worth noting that a hand-tailored solution for the CookingWorld games has been proposed in the "First TextWorld Problems" competition (Côté et al., 2018). This solution[6] managed to obtain up to 91.9% of the maximum possible score across the 514 test games on an unpublished dataset. However, this solution relies on entity extraction and template filling, which we believe limits its potential for generalization. Therefore, this approach should be viewed as complementary rather than competitor to our approach as it could potentially be used as an alternative way of getting promising trajectories.

## 6 CONCLUSION

In this paper we presented a novel methodology for solving text-based games which first extracts high-performing trajectories using phase 1 of Go-Explore and then trains a simple Seq2Seq model that maps observations to actions using the extracted trajectories. Our experiments show promising results in three settings, with improved generalization and sample efficiency compared to existing methods. Finally, we discussed the limitations and possible improvements of our methodology, which leads to new research challenges in text-based games.

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

# A APPENDIX

## A.1 TEXT-GAMES

**CoinCollector** In the *hard* setting (mode 2), each room on the path to the coin has two distractor rooms, and the *level* (e.g. 30) indicates the shortest path from the starting point to the coin room.

**CookingWorld** The game's complexity is determined by the number of skills and the types of skills that an agent needs to master. The skills are:

- **recipe** {**1,2,3**}: number of ingredients in the recipe
- **take** {**1,2,3**}: number of ingredients to find (not already in the inventory)
- **open**: whether containers/doors need to be opened
- **cook**: whether ingredients need to be cooked
- **cut**: whether ingredients need to be cut
- **drop**: whether the inventory has limited capacity
- **go** {**1,6,9,12**}: number of locations

Thus the hardest game would be a recipe with 3 ingredients, which must all be picked up somewhere across 12 locations and then need to be cut and cooked, and to get access to some locations, several doors or objects need to be opened. The handicap of a limited capacity in the inventory makes the game more difficult by requiring the agent to drop an object and later on take it again if needed. The grammar used for the text-based games is the following:

- go, look, examine, inventory, eat, open/close, take/drop, put/insert
- cook X with Y $\longrightarrow$ grilled X (when Y is the BBQ)
- cook X with Y $\longrightarrow$ roasted X (when Y is the oven)
- cook X with Y $\longrightarrow$ fried X (when Y is the stove)
- slice X with Y $\longrightarrow$ sliced X
- chop X with Y $\longrightarrow$ chopped X
- dice X with Y $\longrightarrow$ diced X
- prepare meal
- Where Y is something sharp (e.g. knife).

## A.2 MODELS

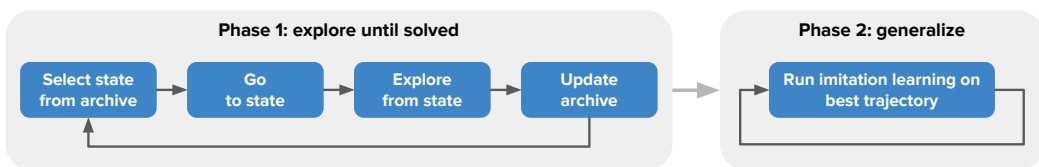

Figure 3: High level intuition of the Go-Exlore algorithm. Figure taken from Ecoffet et al. (2019) with permission.

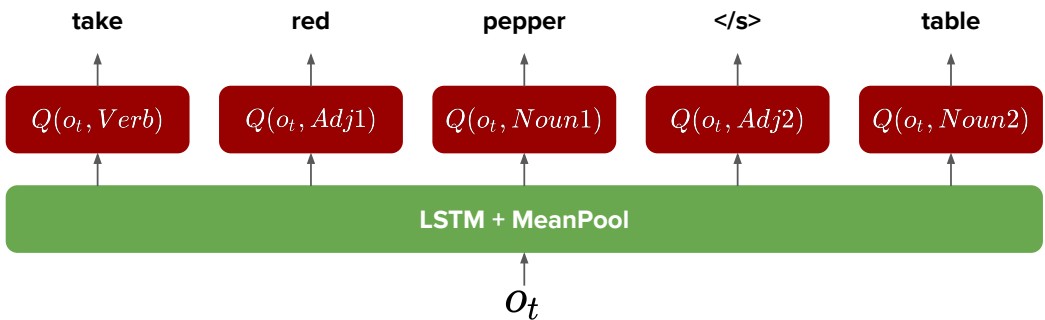

Figure 4: LSTM-DQN high level schema.

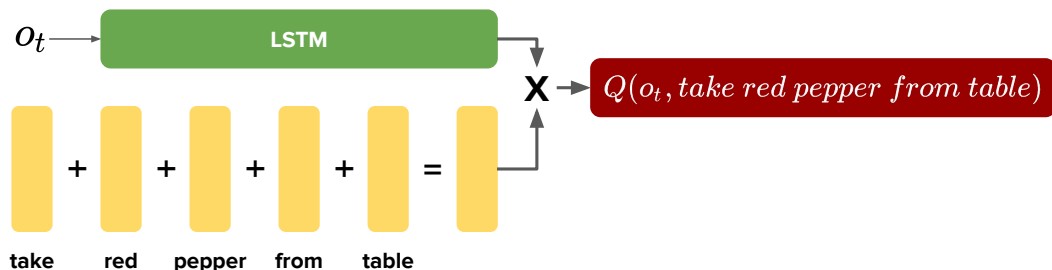

Figure 5: DRRN high level schema.

## A.3 PLOTS

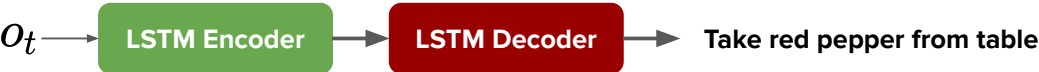

Figure 6: Seq2Seq for imitation learning.

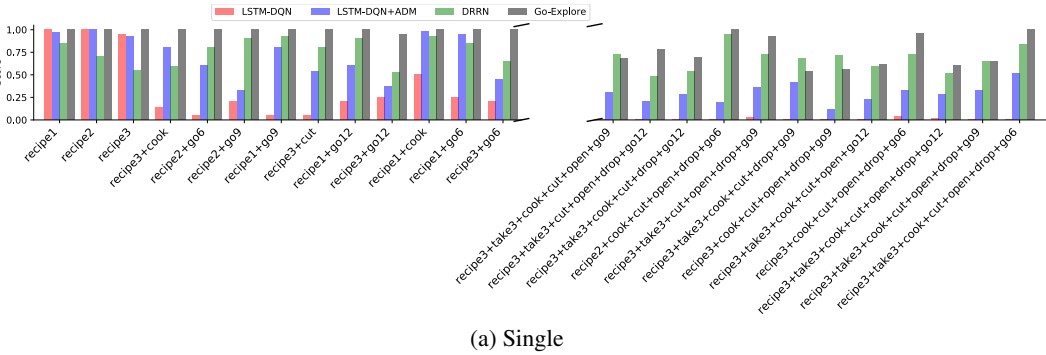

(a) Single

Figure 7: Breakdown of results for the CookingWorld games in the Single setting. The results are normalized and sorted by increasing difficulty level from left to right, averaged among the 20 games of each level.

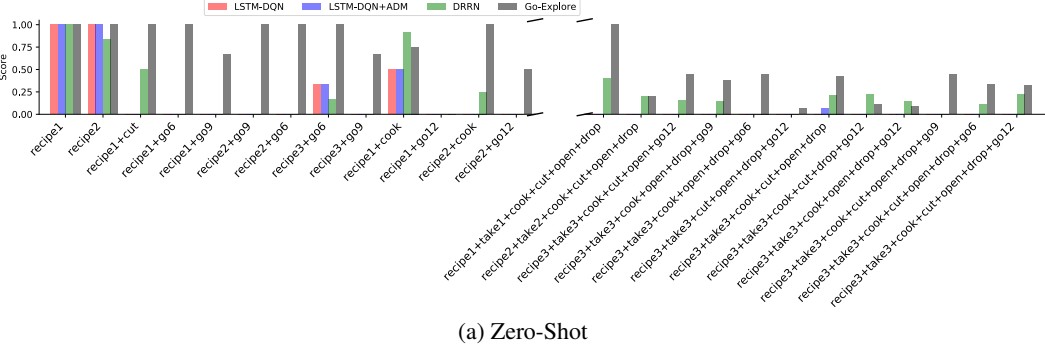

(a) Zero-Shot

Figure 8: Breakdown of results for the CookingWorld games in the Zero-Shot setting. The results are normalized and sorted by increasing difficulty level from left to right, averaged among the 20 games of each level.

Figure 9: Single all the games

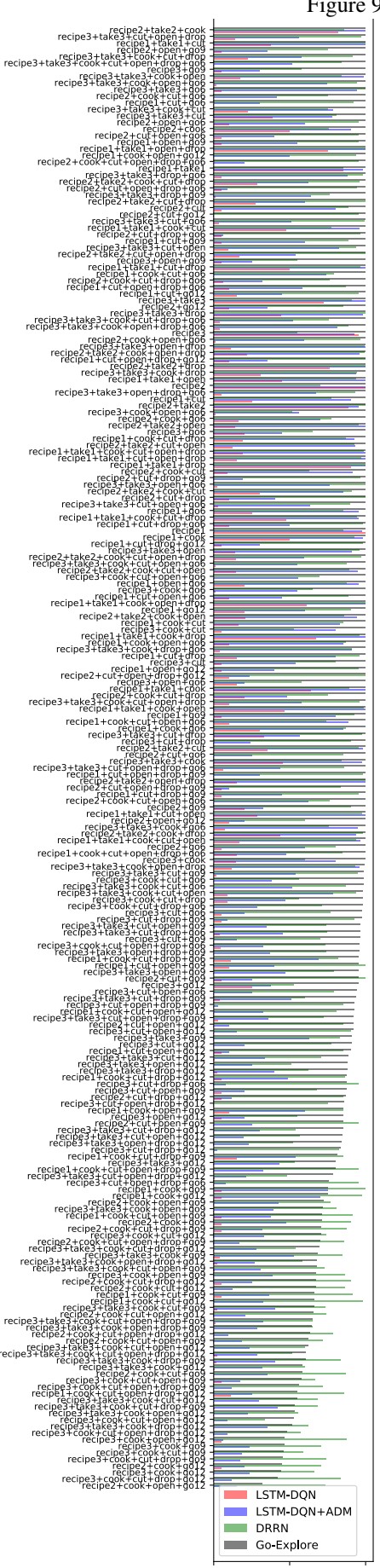

Figure 10: Joint all the games

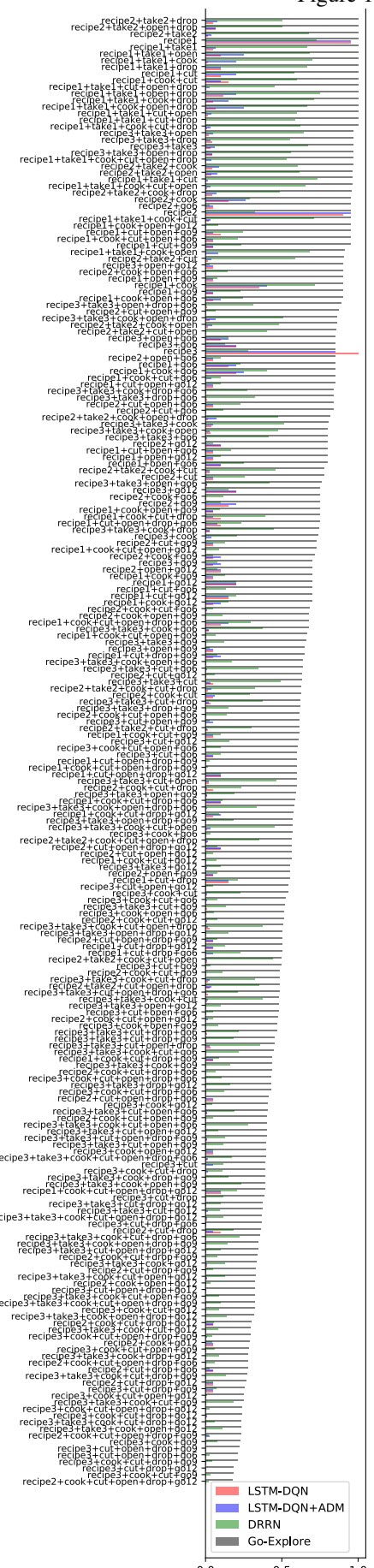

Figure 11: Zero-Shot all the games

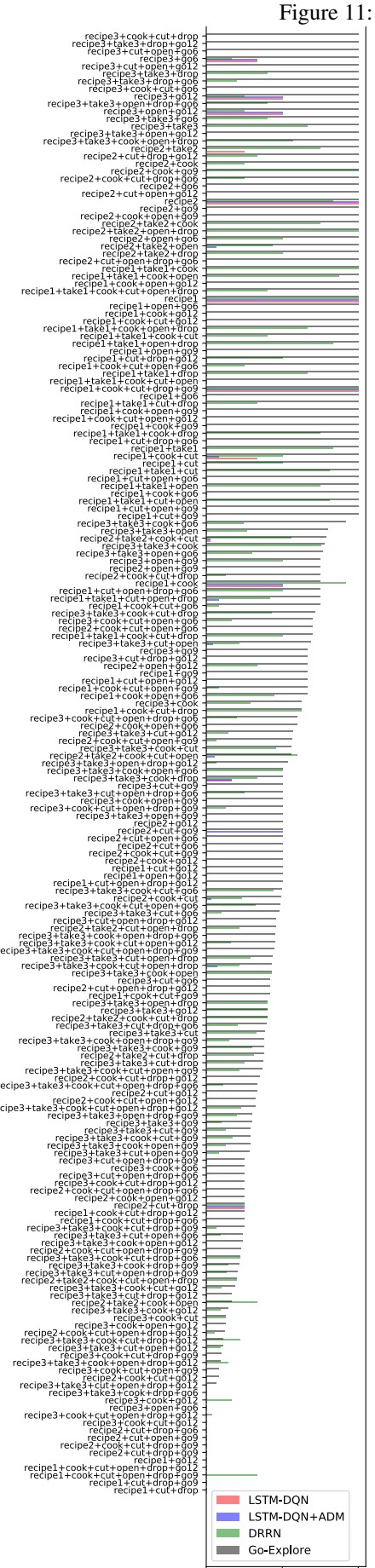

