# OpenReview forum: "Exploration Based Language Learning for Text-Based Games"
_ICLR.cc/2020/Conference — Reject_

### Official Review · AnonReviewer3 · 2019-10-15
**Official Blind Review #3**

**Rating:** 6

**Review:**

This paper applies the Go-Explore algorithm to the domain of text-based games and shows significant performance gains on Textworld's Coin Collector and Cooking sets of games. Additionally, the authors evaluate 3 different paradigms for training agents on (1) single games, (2) jointly on multiple games, and (3) training on a train set of games and testing on a held-out set of games. Results show that Go-Explore's policies outperform prior methods including DRRN and LSTM-DQN. In addition to better asymptotic performance Go-Explore is also more efficient in terms of the number of environment interactions needed to reach a good policy.

Broadly I like how this paper shows the potency of Go-Explore applied to deterministic environments such as the CoinCollector/CookingWorld  games. It is an algorithm that should not be ignored by the text-based game playing community. I also like the fact that this paper clearly explains and demonstrates how efficient and effective Go-Explore can be, particularly when generalizing to unseen games.

The major drawback of the paper is a lack of novelty - the Go-Explore algorithm is already well known, and this paper seems to be a direct application of Go-Explore to text-based games. While the results are both impressive and relevant for the text-game-playing community - it's my feeling that this work may not be of general interest to the broader ICLR community due to the lack of new insights in deep learning / representation discovery. However, I am open to being convinced otherwise.

Minor Comments:

The textworld cooking competition produced at least one highly performing agent (designed by Pedro Lima). While I'm not sure if the code or agent scores are available, it would be a relevant comparison to see how well Go-Explore compared to this agent. (See https://www.microsoft.com/en-us/research/blog/first-textworld-problems-the-competition-using-text-based-games-to-advance-capabilities-of-ai-agents/)

**Experience Assessment:**

I have published one or two papers in this area.

**Review Assessment: Checking Correctness Of Derivations And Theory:**

N/A

**Review Assessment: Checking Correctness Of Experiments:**

I carefully checked the experiments.

**Review Assessment: Thoroughness In Paper Reading:**

I read the paper thoroughly.

---

> ### Author Response · Authors · 2019-11-15
> **Re: Reviewer #3**
>
> Thank you very much for your feedback, let us try to address your concerns,
>
> > Broadly I like how this paper shows the potency of Go-Explore applied to deterministic environments such as the CoinCollector/CookingWorld  games. It is an algorithm that should not be ignored by the text-based game playing community. I also like the fact that this paper clearly explains and demonstrates how efficient and effective Go-Explore can be, particularly when generalizing to unseen games.
>
> Thank you for pointing this out, analyzing the generalization properties of an agent trained with our Go-Explore variant was one of our main goals.
>
> > The major drawback of the paper is a lack of novelty - the Go-Explore algorithm is already well known...
>
> We believe the novelty in this work lies primarily in the fact that it treats language understanding not merely as a reinforcement learning problem, but as a hard-exploration problem. We show that the hard-exploration framing allows us to consider a powerful exploration algorithm, namely Go-Explore, and that the problem thus considered leads to a better language representation than when considered as a classical RL problem, as shown by our high validation-set performance.
>
> Further, although it is true that Go-Explore is now relatively well-known, it is also very recent and to our knowledge, it hasn't yet been shown to work outside of the Atari games Montezuma's Revenge and Pitfall in any published work. Thus it seems to us that our work already has some novelty value from showing Go-Explore's ability to function outside of Atari. Differently from the original Go-Explore method and settings:
> we included the reward in the state representation (which provided a substantial boost in terms of game completion and number of steps to obtain winning trajectories) and we proposed a simple and efficient method for obtaining cell representations from text consisting in word embeddings and binning (which is substantially different from the downscaling of image resolution used in the original go-explore)
> we explicitly test the generalization capabilities of our agents beyond the environment they are trained on, analyzing their generalization capability to new environments, something that was not performed in the original go-explore paper as the agent was both trained and tested on the same Montezuma's Revenge or Pitfall environment. Moreover, the new environment has potentially different action sets from the ones the agents are trained on, which is also a substantial difference with the original Go-Explore and a more challenging scenario where we showed substantial improvement over baselines.
>
> Minor
>
> > The textworld cooking competition produced at least one highly performing agent...
>
> Yes, we actually saw this blog post a few days before the submission, however, no code was provided and we could not implement such a baseline in the time before submission. For the purpose of comparison, the system from the blog post, which was hand-tailored to this environment, managed to obtain up to 91.9% of the maximum possible score across the 514 test games on an unpublished dataset, but relied heavily on entity extraction and template filling, severely limiting its potential of generalizing to other scenarios. Therefore, this approach should be viewed as complementary rather than  competitor  to  our  approach  as  it  could  potentially  be  used  as  an  alternative  way  of  getting promising trajectories. We added this information in the discussion section.

---

### Official Review · AnonReviewer2 · 2019-10-24
**Official Blind Review #2**

**Rating:** 3

**Review:**


This paper considers the task of training an agent to play text-based computer games. One of the key challenges is the high-dimensional action space in these games, which poses a problem for many current methods. The authors propose to learn an LSTM-based decoder to output the action $a_t$ by greedily prediction one word at a time. They achieve this by training a sequence to sequence model on trajectories collected by running the game using a previously proposed exploration method (Go-Explore). While the results are promising, there might be limited novelty beyond training a sequence to sequence model on pre-collected trajectories. Further, the experiments are missing key elements in terms of proper comparison to baselines.

Pros:
1. Nice idea for tackling the unbounded action space problem in text-based games.

Cons:
1. The method depends on the assumption that we can get a set of trajectories with high rewards. This seems a pretty strong assumption. In fact, the authors use a smaller set of admissible actions in order to collect these trajectories in the first place - this seems to not be in line with the goal of solving the large action space problem. If we assume access to this admissible action function, why not just use it directly?
2. Some of the empirical results may not be fair comparisons (unless I'm missing something). For example, all the baselines for the CookingWorld games use $\epsilon$-greedy exploration. Since the Go-Explore method assumes access to extra trajectories at the start, this doesn't seem fair to the other baselines which may not observe the same high-reward trajectories.

Other comments:
1. Do you use the game rewards to train/finetune the seq2seq model or is it only trained in a supervised fashion on the trajectories? (like an imitation learning setup)
2. How critical is the first step of producing high reward trajectories to the overall performance? Some more analysis or discusssion on this would be helpful to disentangle the contribution of GoExplore from the seq2seq action decoder.

**Experience Assessment:**

I have published one or two papers in this area.

**Review Assessment: Checking Correctness Of Derivations And Theory:**

N/A

**Review Assessment: Checking Correctness Of Experiments:**

I assessed the sensibility of the experiments.

**Review Assessment: Thoroughness In Paper Reading:**

I read the paper at least twice and used my best judgement in assessing the paper.

---

> ### Author Response · Authors · 2019-11-15
> **Re: Review #2**
>
> Thanks for your comments, let us try to address your valid concerns:
>
> > Nice idea for tackling the unbounded action space problem in text-based games
>
> Thanks for pointing this out, it is indeed one of the main problems we tried to tackle in the paper and a problem we deeply care about.
>
> Cons:
>
> > 1. The method depends on the assumption that we can get a set of trajectories with high rewards...
>
> Thank you for your feedback. For these experiments, we did indeed assume that admissible action information is available in our training environment, but the eventual goal is to generalize to test environments where such information is not available. The assumption that admissible actions are available at training time holds in cases where we build the training environment for the RL agent (e.g. a hand-crafted dialogue system), and a system trained in such an environment can be practically applied as long as the system does not rely on such information at test time. Thus, we assumed that these admissible commands are not available at test time.
> In addition, our experiments demonstrate that even when admissible commands are available, learning how to rank them is not easy either (DRRN baseline), especially in unseen games (zero-shot settings). In the discussion section (i.e. Language-Based Exploration) we explore the limitations of using the admissible action during Phase 1 and how in general a Language Model could help to generate admissible action on the fly. The reported results, to the best of our knowledge, are the first of this kind in Text-based games.
>
> > 2. Some of the empirical results may not be fair comparisons...
>
> This is a good point, as we mentioned to Reviewer 1, in In Cooking-World we also tried DQN with the same counting based reward, but then this was hurting the overall performance in Single setting and especially in the Joint setting. We will update the paper to include these additional results that provide a more fair comparison. On the other hand, we tried to address this point with the experiments in CoinCollector, a game with very sparse reward, by using two baselines with count-based exploration rewards. In this setting, Go-Explore exploration resulted to be more sample efficient as you also pointed out in the pros.
>
> Comments
>
> > 1. Do you use the game rewards to train/finetune the seq2seq model...
>
> In our current work, we used imitation learning without using rewards directly, but in general, we could further finetune the policy using RL. We opted for a simpler solution since the results were already promising, we will add a further discussion about this in the paper. We edited the paper to clarified this explicitly.
>
> > 2. How critical is the first step of producing high reward trajectories to the overall performance? ...
>
> Thank you, this is a good suggestion that would add additional depth to our analysis. For example, we could try to train the Seq2Seq model with the second or third best trajectory found by phase1 and report the results for the three settings. Since we used pure imitation learning, we expect better trajectories to lead to a better policy. We will include these experiments in the next version of the paper.

---

> > ### Comment · AnonReviewer2 · 2019-11-15
> > **thanks**
> >
> > Thank you for the response. I believe exploring some of these steps would add more depth to the experiments as well as the analysis and make the paper stronger in the next revision!
> >
> > Also, to clarify the point I made in Con 1 a bit: I meant that the assumption of getting **trajectories with high rewards** is pretty strong since that is part of the exploration problem in the first place.

---

> > > ### Author Response · Authors · 2019-11-15
> > > **Further answer**
> > >
> > > Thank you for your quick response!
> > >
> > > We do agree that getting trajectories with high rewards is a strong assumption in general; however the empirical results demonstrate that, given a restricted action space like in our TextWorld setting, Go-Explore is effective at obtaining these high-scoring trajectories.
> > >
> > > In the case of text games, a restricted action space means assuming access to admissible actions, but as we discuss, we believe this assumption is met in many useful cases, like dialogue systems, and we also suggest the possibility of obtaining admissible actions using a general language model.
> > >
> > > In general, as long as the Go-Explore algorithm is able to efficiently explore a search space, high-scoring trajectories can be obtained. It is an open question to what degree Go-Explore will scale as environments get more complicated, but we believe that, given appropriate priors on actions and a sufficiently informative cell representation, Go-Explore should be able to scale to environments that are much more complex than those presented in this paper. That said, testing the limits of Go-Explore is a direction for future work.

---

### Official Review · AnonReviewer1 · 2019-10-26
**Official Blind Review #1**

**Rating:** 3

**Review:**

This paper proposes an exploration approach of Go-Explore together with imitation learning for playing text games. It is shown to outperform existing solutions in solving text-based games with better sample efficiency and stronger generalization ability to unseen games.

Pros:
Seq2seq imitation learning + Go-Explore is applied to more challenging text games and achieves better performance, higher sample complexity and better generalization ability.

Cons:
•	From modeling perspective, the policy network uses the standard sequence-to-sequence network with attention. And it is trained on the high-reward trajectories obtained with Go-Explore method using imitation learning. From this perspective, there is not much novelty in this paper.

Detailed comments:
•	More details about the mapping function f(x) in Phase 1 should be given.
•	It is not clear why Phase 2 should be called “Robustification”. It seems to be just standard imitation learning of seq2seq model on the high-reward trajectories collected in Phase 1.
•	In the paragraph after eqn. (1), H is defined to be the hidden states of the decoder. Shouldn’t it be the hidden states of the encoder?
•	It seems to be unfair to compare the proposed method with advanced exploration strategy to other model-free baselines that only have very simple exploration strategies (e.g., epsilon-greedy). It is not surprising at all that Go-Explore should outperform them on sparse reward problems. More baselines with better exploration strategies should be compared.

**Experience Assessment:**

I have published in this field for several years.

**Review Assessment: Checking Correctness Of Derivations And Theory:**

I carefully checked the derivations and theory.

**Review Assessment: Checking Correctness Of Experiments:**

I carefully checked the experiments.

**Review Assessment: Thoroughness In Paper Reading:**

I read the paper thoroughly.

---

> ### Author Response · Authors · 2019-11-15
> **Re: Review #1**
>
> Thank you very much for your comments.
>
> > Seq2seq imitation learning + Go-Explore is applied to more challenging text games and achieves better performance, higher sample complexity and better generalization ability.
>
> The performance of our Go-Explore variant in indeed one of the main strenghts of the paper, we are glad you appreciated this aspect.
>
> > Cons: From modeling perspective...
>
> We purposely decided to use a simple and general model for the policy to keep our methodology applicable to any kind of text-based game or for example to other interactive textual application (e.g. dialogue systems). Our work is actually the first (that we are aware of) to use Seq2Seq models in this domain, as most literature relies on an architecture with a fixed number of Q-value heads, one for each of the action tokens (e.g. DQN). In preliminary experiments, we also tried more complex architectures like transformers, but the performance gain was minimal, so we decided to keep the model as simple as possible. We will add the additional more complex policies in the coming version of the paper (e.g. Transformers) for reference and for showing the appropriateness of our explicit choice of a simple architecture.
>
> In addition, we believe that separating exploration and exploitation (in our case using Go-Explore) is a novel approach in text-based games, is novel in text-based games, which are very different in nature compared to usual RL benchmarks (e.g. Atari). Indeed, this method outperforms existing baselines and allows for deploying a less constrained policy model (e.g. Seq2Seq). Being able to use such models --Seq2Seq-- in Textworld allows for more complex games where there is no known/predefined sentence structure as required by DQN models.
>
> Detailed comments:
>
> > More details about the mapping function...
>
> Thank you for indicating that some details of the mapping function were missing in the initial version of the paper, we will add more details about the phase 1 mapping function f(x). Specifically, we changed the text to clarify that f(x) is the average on the word-embedding of the observation and added further clarification on the effects of including the reward.
>
> > It is not clear why Phase 2 should be called “Robustification”
>
> We followed the terminology from the Go-Explore paper by referring to phase 2 as the robustification phase. Their rationale for this term is that this phase turns a fragile policy of playing a trajectory of actions in sequence into a more robust, state-conditioned policy that can thus deal with environmental stochasticity. That said, in our experiments, the purpose of the second phase is a generalization and, as you correctly pointed out, such a general policy is obtained through standard imitation learning. We updated the manuscript accordingly, renaming the phase to make it more clear to the reader and clarifying the correspondence between the original go-explore robustification phase with our generalization phase.
>
> > In the paragraph after eqn. (1)
>
> Yes, that was a typo, we corrected that.
>
> > It seems to be unfair to compare the proposed method with...
>
> This is a good point. In Cooking-World we also tried DQN with the same counting based reward, but this was hurting the overall performance in the Single setting and especially in the Joint setting. We will update the paper to include these additional results that provide a fairer comparison. On the other hand, we tried to address this point with the experiments in CoinCollector, a game with very sparse reward, by using two baselines with count-based exploration rewards. In this setting, Go-Explore exploration was more sample efficient, as you also pointed out in the pros.

---

### Decision · Program_Chairs · 2019-12-19

**Decision:**

Reject

**Comment:**

The paper applies the Go-Explore algorithm to text-based games and shows that it is able to solve text-based game with better sample efficiency and generalization than some alternatives.  The Go-Explore algorithm is used to extract high reward trajectories that can be used to train a policy using a seq2seq model that maps observations to actions.

Paper received 1 weak accept and 2 weak rejects.  Initially the paper received three weak rejects, with the author response and revision convincing one reviewer to increase their score to a weak accept.

Overall, the authors liked the paper and thought that it was well-written with good experiments.
However, there is concern that the paper lacks technical novelty and would not be of interest to the broader ICLR community (beyond those that are interested in text-based games).  Another concern reviewers expressed was that the proposed method was only compared against baselines with simple exploration strategies and that baselines with more advanced exploration strategies should be included.

The AC agrees with above concerns and encourage the authors to improve their paper based on the reviewer feedback, and to consider resubmitting to a venue that is more focused on text-based games (perhaps an NLP conference).